# Rhabdomyosarcoma Associated with Core Myopathy/Malignant Hyperthermia: Combined Effect of Germline Variants in *RYR1* and *ASPSCR1* May Play a Role

**DOI:** 10.3390/genes14071360

**Published:** 2023-06-27

**Authors:** Pamela V. Andrade, Joilson M. Santos, Anne C. B. Teixeira, Vanessa F. Sogari, Michelle S. Almeida, Fabiano M. Callegari, Ana C. V. Krepischi, Acary S. B. Oliveira, Mariz Vainzof, Helga C. A. Silva

**Affiliations:** 1Malignant Hyperthermia Unit—Discipline of Anesthesiology, Pain and Intensive Care—Federal University of Sao Paulo, São Paulo 04024-002, SP, Brazil; pamvieira@msn.com (P.V.A.); joilson.dcir@epm.br (J.M.S.); 2Department of Genetics and Evolutionary Biology, Institute of Biosciences, University of São Paulo, São Paulo 05508-090, SP, Brazil; annecarol7@gmail.com (A.C.B.T.); ana.krepischi@ib.usp.br (A.C.V.K.); 3Department of Pathology, Federal University of Sao Paulo, São Paulo 04024-002, SP, Brazil; vane_fante@yahoo.com.br (V.F.S.); fabcalleg@icloud.com (F.M.C.); 4Department of Oncology, Federal University of Sao Paulo, São Paulo 04024-002, SP, Brazil; msoncologia@hotmail.com; 5Department of Neurology, Federal University of Sao Paulo, São Paulo 04024-002, SP, Brazil; acary.bulle@unifesp.br

**Keywords:** malignant hyperthermia, rhabdomyosarcoma, central core disease, ryanodine receptor, ASPSCR1 protein

## Abstract

Rhabdomyosarcomas have been described in association with thyroid disease, dermatomyositis, Duchenne muscular dystrophy, and in muscular dystrophy models but not in patients with ryanodine receptor-1 gene (*RYR1*) pathogenic variants. We described here an 18-year-old male who reported a cervical nodule. Magnetic resonance images revealed a mass in the ethmoidal sinus corresponding to rhabdomyosarcoma. As his father died from malignant hyperthermia (MH), an in vitro contracture test was conducted and was positive for MH susceptibility. Muscle histopathological analysis in the biopsy showed the presence of cores. Molecular analysis using NGS sequencing identified germline variants in the *RYR1* and *ASPSCR1* (alveolar soft part sarcoma) genes. This report expands the spectrum of diseases associated with rhabdomyosarcomas and a possible differential diagnosis of soft tissue tumors in patients with *RYR1* variants.

## 1. Introduction

Rhabdomyosarcoma is a soft tissue sarcoma with myogenic differentiation and an incidence of 4.5 per million in patients under 20 years [1,2]. The alveolar rhabdomyosarcoma histopathological and genetic variant exhibits frequent metastasis, worse prognosis, and the acquisition of somatic balanced chromosomal translocations with novel fusion genes (paired-box transcription factor 3 *(PAX3)* or 7 (*PAX7*) and forkhead box protein O1 (*FOXO1*)) [1,2]. Rhabdomyosarcomas have been described in association with concomitant thyroid disease, dermatomyositis, Duchenne muscular dystrophy, and muscular dystrophy animal models [1,3,4], but have never been reported as associated with ryanodine receptor 1 gene (*RYR1*) [5] pathogenic variants.

*RYR1* encodes the sarcoplasmic reticulum (SR) calcium release channel (or ryanodine receptor—RyR1) and its mutations are associated with susceptibility to malignant hyperthermia (MH [MIM: 145600]) and myopathies, especially congenital central core disease (CCD [MIM 117000]) [5]. MH is an autosomal dominant hypermetabolic syndrome arising after anesthesia with halogenated anesthetics or succinylcholine [5]. While the incidence of MH crisis ranges from 1:3000 to 1:50,000 anesthesias, MH-related variants in *RYR1* occur in 1:2750 individuals [6]. MH crisis probability is calculated with a clinical grade scale, and MH susceptibility can be diagnosed using the in vitro muscle contracture test (IVCT) in response to caffeine and halothane [5,6]. CCD is characterized by areas without oxidative activity (cores) in skeletal muscle fibers and is inherited in both dominant and recessive patterns with variable expressivity, from floppy babies to adults with mild weakness [5]. Different *RYR1* mutations can lead to channel loss of function (myopathy without MH) or channel gain of function (isolated MH or MH with associated myopathy), and both dominant and recessive inheritances have been described [5,7].

The ryanodine receptor protein has three isoforms, RyR1, Ry2, and RyR3, more expressed in the skeletal muscle, heart, and brain, respectively [8]. All three isoforms play a crucial role in intracellular calcium dynamics, regulating physiologic functions such as the skeletal/cardiac muscle excitation–contraction coupling, neuronal excitability, hormone secretion, gene transcription and cell differentiation, migration, and apoptosis [8,9,10]. While *RYR2* mutations have been associated with arrhythmias and heart failure, *RYR3* mutations have been described in association with cardiac and neurodegenerative diseases [8]. Additionally, the calcium signaling pathway has been increasingly linked to cancer development and progression, with the participation of the three isoforms of the ryanodine receptor [8,10]. Recently, transcriptome analysis on skin cancer microcystic adnexal carcinoma revealed the upregulation of four genes of the calcium signaling pathway, including *RYR1*, which was confirmed by increased expression on immunohistochemistry [10].

Here, we report the case of a patient with an *RYR1* pathogenic variant associated with MH and cores who developed an alveolar rhabdomyosarcoma associated with a germline variant in the cancer predisposition gene *ASPSCR1* (Alveolar Soft Part Sarcoma Chromosomal Region Candidate 1). While pathogenic variants in the *RYR1* gene associated with MH/CCD can induce cycles of muscular necrosis and regeneration, variants in the *ASPSCR1* originate an unchecked cellular proliferation: the combined effect of these two variants could have played a role in the development of the rhabdomyosarcoma in this patient [11].

## 2. Materials and Methods

The propositus was evaluated with radiologic studies, tumor and muscle biopsies, the in vitro contracture test (IVCT), and genetic studies. The radiologic study included a spine X-ray, cervical ultrasonography, face and neck computerized tomography (volumetric acquisition was performed in equipment with multidetectors with the intravenous injection of iodinated contrast), and magnetic resonance images (sequences based on spin-echo, gradient-echo or susceptibility-weighted imaging, and planar-echo (diffusion) image acquisitions in the various orthogonal planes, with infusion of paramagnetic contrast medium).

The muscle biopsy was performed under regional anesthesia in the left vastus lateralis muscle. Muscle strips were used for the anatomopathologic study with routine stains (haematoxylin-eosin, Gomori trichrome, Sudan, and periodic acid Schiff), histochemistry (nicotinamide adenine dinucleotide dehydrogenase (NADH), succinate dehydrogenase (SDH), cytochrome *c* oxidase (COX), and myofibrillar adenosine triphosphatase (ATPase) with alkaline (pH 9.4) and acid (pH 4.3) preincubation), and the IVCT, performed as previously described [12]. Briefly, for the IVCT, the muscle fragments were placed in a carboxygenated Krebs-Ringer solution and dissected into thinner fragments (2–3 mm). Next, each fragment was attached to a force transducer (Narco Bio Systems transducer, International Biomedical, City, Houston, TX, USA) inside vats heated to 37 °C, perfused intermittently with Krebs-Ringer’s solution, and oxygenated continuously with carbogen. Each fragment was continuously stimulated with supramaximal electrical impulses of 1 ms and 0.2 Hz (Boeck stimulator) under 2 g of load. The degree of muscle contraction throughout the test was monitored in real time using an analogical–digital amplifier/transformer and the AQDados biological data acquisition program (Links, São Paulo, SP, Brazil). Four cumulative static tests were performed, two with caffeine (0.5; 1; 1.5; 2; 3; 4; and 32 mMol/L) and two with halothane (0.5; 1; 2; and 3%). The test result was referred to as the threshold, which is defined as the lowest concentration that produces a minimum increase of 0.2 g in baseline tension.

The nodule biopsy was performed with fine-needle aspiration cytology under local anesthesia. Cells were analyzed after Giemsa and Papanicolaou stains and an immunohistochemistry panel including antibodies against desmin, myogenin, HHF-35 (Actin, Muscle Specific), Ki-67 (nonhistone nuclear protein, marker of cell proliferation), Epstein Barr virus, enolase, chromogranin, synaptophysin, bcl-2 (B-cell lymphoma-2 apoptosis regulator), bcl-6 (B-cell lymphoma-2 apoptosis regulator), TdT (terminal deoxynucleotidyl transferase), AE1 (band 3 anion transport protein), AE3 (anion exchange protein 3), CD3 (cluster of differentiation 3 B-cell protein complex), CD10 (cluster of differentiation 10 B-cell protein complex or common acute lymphocytic leukemia antigen), CD20 (cluster of differentiation 20 B-cell protein complex), and CD99 (cluster of differentiation 99 B-cell protein complex or single-chain type-1 glycoprotein).

Genetic studies: Chromosome microarray analysis (CMA) for copy number variant investigation was performed using genomic DNA extracted from peripheral blood using a 180 K platform (Agilent Technologies, Santa Clara, CA, USA) according to the manufacturer’s recommendations, and the analysis was conducted as previously described [13]. Germline whole-exome sequencing was performed with blood DNA to construct genomic libraries using the Sure Select Human All Exon V6 kit (Agilent Technologies), with resulting quality control metrics of 117X median coverage, 99.6% on target at >10x, 84% > Q30. Variant annotation was carried out through Varstation software (https://varstation.com/, accessed on 10 June 2023), and filtering was based on allele frequency (≥0.35), read depth (≥20), genotype quality (Phred score ≥ 20), population frequencies ≤ 1% (1Kgenomes, gnomAD, and ABraOM), and nonsynonymous coding substitutions (missense/loss of function). Variants were prioritized using clinical databases (OMIM, ClinVar, and HGMD). All variants mapped to OMIM morbid genes, including cancer predisposition genes, were evaluated. Clinically relevant variants were classified according to the ACMG criteria [14].

The ethics committee approved this study under the number 73681017.9.0000.5505, and all family members provided written informed consent according to the Declaration of Helsinki.

## 3. Results

### 3.1. Clinical Description of the Patient

An 18-year-old male had a cervical nodule for 1 month. His mother reported decreased fetal movement during his gestation, and he was delivered prematurely at 7 months. He presented a subsequent normal psychomotor development. He has been followed due to sleep apnea and snoring since childhood and the patient reported a lateral deviation of the spine for the last 2 years. His father died from MH after thyroidectomy surgery (see Section 3.2 for details of the father’s malignant hyperthermia crisis). The clinical grading scale scored the crisis risk at 58 (almost certain risk for MH) [6]. One patient’s paternal cousin had ovarian cancer at 14 years of age.

Examination of the patient revealed a 5 cm cervical nodule, facial dysmorphisms (strabismus, eyelid ptosis, micrognathia, everted ears, high arched palate), global muscle hypotrophy, joint hypermobility, hypoactive reflexes (upper limbs), and scoliosis (positive Adam’s test; spine X-ray: scoliosis with Cob’s angle of 29°) (Figure 1a). His gait was normal and Gower’s sign was negative; his muscle strength was 5/5 according to the Medical Research Council grading system. He sustained outstretched arms and the lower limb Mingazzini maneuver for two minutes. His serum creatine kinase (CK) level was increased (621 IU/L; maximum normal value of 308 IU/L for men), the electrocardiogram (ECG) was normal, and the forced vital capacity was 90% of the predicted value. His IVCT was positive for MH susceptibility, with contractures of 2.28 g at 2% halothane and 2.12 g at 2 mMol caffeine (normal threshold 0.2 g) (Figure 1b,c).

### 3.2. Father’s Malignant Hyperthermia Crisis Description

A male patient in his 30s presented with a two-year complaint of weight loss, cervical enlargement, hot flashes, sweating, and eye irritation. On examination, there was thyroid enlargement of twice the normal size as well as congenital pectus carinatum, micrognathia, strabismus, and eyelid ptosis. The patient was diagnosed with Basedow-Graves disease and medicated with propylthiouracil 1100 mg, with partial improvement. Total thyroidectomy surgery under general anesthesia was indicated, and lugol was administered four days before the surgery. The patient was monitored with an electrocardiogram, pulse oximetry, noninvasive blood pressure, and bladder catheter. He was premedicated with midazolam 2 mg, and anesthetic induction was performed with fentanyl 150 mcg, propofol 170 mg, and atracurium 50 mg. After endotracheal intubation, anesthetic maintenance was performed with halothane and nitrous oxide. After two hours of surgery, the patient presented hypercarbia (80 mmHg ETCO_2_), hyperthermia (43.7 °C, oropharyngeal), tachycardia (135 bpm), hypotension (100 × 60 to 80 × 30 mmHg), and a decrease in peripheral oxygen saturation to 96%. Halothane was discontinued, and malignant hyperthermia crisis treatment was initiated with dantrolene, cooling measures (intravenous iced solutions, bladder, and gastric lavage with iced solutions, cooling blanket, and placement of ice packs on the body surface), metabolic acidosis treatment (sodium bicarbonate), hyperkalemia management (calcium gluconate, glucose/insulin solution), procainamide, and dexamethasone. He was manually ventilated to allow the change to an anesthetic machine with a new soda lime and circuit. The patient had two cardiorespiratory arrests, which were reversed (at 1:30 and 3 h from the beginning of the malignant hyperthermia crisis). He was transferred to the intensive care unit, where he developed shock after persistent bleeding by disseminated intravascular coagulation, with death 20 h after the onset of the crisis.

### 3.3. Muscle Biopsy Analysis

The muscle biopsy analysis showed increased variability in fiber size, hypertrophic and atrophic fibers, splitting, increased nuclear centralization (29%), fibrosis (endomysium and perimysium), type I fiber predominance (70%), and cores in 9% of type I muscle fibers (Figure 1d). There was no accumulation of glycogen or fat nor negative Cox muscle fibers.

Cervical ultrasonography revealed an oval isolated hypoechogenic solid nodule (4.1 × 2.9 × 1.9 cm). Face and neck magnetic resonance images (Figure 1e,f) revealed a solid infiltrative heterogeneous mass (4.4 × 6.5 cm) with ill-defined boundaries, infiltrative aspect, and heterogeneous contrast enhancement, whose geometric center was located in the ethmoidal sinus, determining remodeling and erosion of its cells and ethmoid perpendicular lamina. There was an extension to the left nasal bone and cavity, frontal/sphenoid/jaw sinuses, and orbital cavity (no cleavage plane with the medial rectus muscle), with intracranial area occupation (olfactory gutters) and compression of adjacent frontobasal parenchyma beyond submandibular/cervical/retropharyngeal lymph node enlargement.

### 3.4. Histopathology of the Tumor

Fine-needle aspiration cytology of the nodule revealed metastasis of alveolar rhabdomyosarcoma (Figure 2). There were isolated and grouped small, rounded cells (Figure 2A), with a scarce or micro-vacuolated cytoplasm (Figure 2B), oval, regular, and normochromic nuclei with multiple small nucleoli, mitosis (Figure 2C), apoptotic bodies (Figure 2D), and multinucleated giant tumor cells (Figure 2E). Immunohistochemistry was positive for the markers of myogenic differentiation myogenin (Figure 2F), desmin (focal labeling—Appendix A), and HHF-35, with a positive Ki-67 immunolabeling in 50% of cells. Immunohistochemistry was negative for the other antibodies tested (Epstein Barr virus, enolase, chromogranin, synaptophysin, bcl-2, bcl-6, TdT, AE1, AE3, CD3, CD10, CD20, and CD99).

The modified clinicopathological staging (TNM classification) for this patient was initially three (site of primary tumor T2b, involvement of regional lymph nodes N1, distant metastatic spread M0). The patient underwent radiotherapy (5 fractions of 4000 Gy) and chemotherapy with vincristine, dactinomycin, and cyclophosphamide (3 cycles). It was modified to carboplatin and etoposide (two cycles) after disease progression (enlargement of the primary lesion and cervical lymph node, extending to the supraclavicular chain). After the detection of hip bone metastases, the chemotherapy regimen was changed to vinorelbine (two cycles); however, the patient died after 10 months.

### 3.5. Germline Genetic Studies: Disclosing a RYR1 Pathogenic Variant and a VUS in ASPSCR1

The patient’s G-banded karyotype produced a normal result. Whole-exome sequencing was performed and all rare coding nonsynonymous/splicing variants mapped to OMIM morbid genes were evaluated.

The analysis detected a known pathogenic variant in heterozygosity mapped to exon 46 of the *RYR1 gene* (RefSeq NM_000540.2(RYR1):c.7354C>T (p. Arg2452Trp)) (Figure 3, left). This variant was previously described in association with MH/CCD (ClinVar RCV000056226.1) [15,16]. No other clinically relevant variant was detected in the *RYR1* gene nor in other genes known to be related to MH (*CACNa1S, STAC3,* and *TRPV1*).

In addition, in spite of the facial dysmorphisms, additional clinically relevant variants in other OMIM morbid genes were not detected.

Subsequently, using an in-house list of cancer predisposition genes for annotation, variants mapped to all known and candidate genes were evaluated. Pathogenic/likely pathogenic variants mapped to known cancer predisposition genes were not detected in the patient. In particular, we found 14 rare germline heterozygous missense variants (frequency < 0.05%) mapped to candidate cancer predisposition genes, including a missense variant in exon 6 of the *ASPSCR1* gene (Figure 3, right), which was classified as a variant of unknown significance (VUS), according to the ACMG 2015 criteria [14] (RefSeq NM_001251888.1(ASPSCR1):c.449C>A (p.Ala150Asp)). This *ASPSCR1* VUS is likely to have a damaging effect in the protein function according to the in silico analysis (aggregated prediction using methods from Ensemble REVEL and MetaLR).

## 4. Discussion

We presented the case of a patient with MH, cores, and alveolar rhabdomyosarcoma with a germline *RYR1* pathogenic variant and a germline VUS in the *ASPSCR1* gene. The association of rhabdomyosarcoma with muscular dystrophy has been related to the loss of proteins with protective action (DNA repair and antitumor activity) in dystrophic muscle; additionally, cycles of necrosis–regeneration–inflammation–fibrosis, common to many myopathies, could predispose patients to tumor development [1]. While alveolar rhabdomyosarcoma mainly affects the trunk and extremities, our patient had a periorbital mass. We speculated that the myopathic involvement of extraocular muscles could have contributed to the expression of oncogenic fusion proteins (*PAX3–FOXO1*, *PAX7*, and *FOXO1*) in periorbital skeletal myoblasts during necrosis–regeneration cycles [1].

Additionally, there were diagnostic challenges related to the differential diagnosis between MH and thyrotoxicosis in the patient’s father [5,6] and the patient’s unusual phenotype of juvenile scoliosis without muscle weakness. In patients with *RYR1* gene gain-of-function mutations, the presence of triggering anesthetics leads to excessive calcium release from the sarcoplasmic reticulum, which causes muscle stiffness and hypermetabolism characterized by increased oxygen uptake and carbon dioxide and lactic acid production [5,6]. Then, a typical malignant hyperthermia crisis is characterized by tachycardia, tachypnoea, hypercarbia, hyperthermia, muscle rigidity, acidosis, rhabdomyolysis, hyperkalemia with arrhythmias, myoglobinuria with secondary renal failure, compartment syndrome, disseminated intravascular coagulation, pulmonary edema, and even central nervous system injury [5,6]. Thyrotoxicosis during surgical manipulation is a differential diagnosis of MH. Both present with tachycardia, tachypnea, and hyperthermia; however, typically only MH presents with hypercarbia, muscle rigidity, acidosis/hyperkalemia, and rhabdomyolysis due to excessive calcium release from the SR [5,6,17,18].

Regarding scoliosis, predominant or exclusive axial myopathy of late onset is a presentation of *RYR1* gene pathogenic variants and can be present with or without cores on muscle biopsy [19]. Even though the proband did not carry additional variants in genes related to MH, he had some features that could be found in recessive *RYR1* mutations, such as facial dysmorphism, strabismus, ptosis, joint hyperlaxity, muscle hypotrophy, and increased nuclear centralization. Previous reports of the pathogenic variant c.7354C>T in *RYR1* described a variable pattern of dysmorphic features and muscle biopsy (ClinVar RCV000056226.1) that ranged from MH with CCD to MH with cleft palate, increased creatine kinase, a strongly positive in vitro muscle contracture test, and normal muscle histology [15,16].

Genetic predisposition was detected in only 7–10% of rhabdomyosarcoma patients [20]; in the present case, mutations in known cancer susceptibility genes could be excluded, but notably, a missense VUS was detected in *ASPSCR1*, and such variants have been associated with glioma and leukemia in pediatric patients [20]. Some cases of alveolar soft tissue sarcomas have been found to be caused by the fusion of *ASPSCR1* and *TFE3* [21]. In addition, it was recently shown that in alveolar soft part sarcoma, the expression of the fusion transcription factor *ASPSCR1::TFE3* is required for in vivo tumor development via angiogenesis, orchestrating higher-ordered angiogenesis via super-enhancers activity [22]. The protein encoded by *ASPSCR1* belongs to the family of UBX domain-containing proteins and regulates the trafficking of GLUT4 glucose transporters in adipocytes and muscle cells [23,24]. Because the Golgi multicompartment stack structure provides an optimal setting for glycosylation, dysregulation of the Golgi structure and Golgi structural proteins has been highly related to glycosylation defects [25]. The role of the Golgi complex and glycosylation defects in cancer has recently been highlighted [26]. Therefore, it is legitimate to speculate that a germline deleterious variant in genes causing Golgi dysregulation, such as *ASPSCR1*, may lead to altered glycosylation and possibly contribute to tumorigenesis in cells expressing such mutant or detective protein, such as adipocytes and muscle cells. Taken together, these two pieces of evidence imply a possible role for the detected germline *ASPSCR1* variant in rhabdomyosarcoma development.

Another relevant issue to be considered is the role of the germline *RYR1* variant in the cancer development in this case. Abnormalities in Ca^2+^ signaling are frequently observed in tumors, representing common alterations [27]. The remodeling of Ca^2+^ signaling in cancer can have significant impacts on crucial processes such as proliferation, invasion, and cellular susceptibility to death. Among the various types of Ca^2+^ release channels, one notable example is RyR, and the mutational landscape of the three RyR isoforms had not been systematically investigated until a recent study by Wang et al. [28]. In this study [28], they evaluate the mutational landscape of RyR and its correlation with various cancer phenotypes, utilizing data from the TCGA database. Their findings unveiled characteristic RyR mutational patterns in cancers and established correlations with tumor mutational burden (TMB), patient prognosis, age, and smoking status; however, rhabdomyosarcomas were not evaluated. So far, there is no reported link between *RYR1* gene mutations and rhabdomyosarcoma development. However, similar to dystrophin deficiency, increased intracellular calcium, a characteristic of many myopathies [1], could theoretically be a contributing factor in patients who already carry variants in genes that may increase rhabdomyosarcoma susceptibility. SR calcium-ATPase (*SERCA*) is expressed in rhabdomyosarcomas, suggesting that intracellular calcium could participate in tumor cell proliferation [29]. Additionally, CD97 (an adhesion class G protein-coupled receptor), which is expressed at the peripheral sarcolemma and colocalizes with *SERCA*, is strongly upregulated in rhabdomyosarcomas compared to normal skeletal muscle [30] and could be involved in myogenic differentiation of sarcomas.

## 5. Conclusions

This report expands the spectrum of diseases reported with rhabdomyosarcomas and illustrates that the occurrence of a soft tissue tumor in patients with *RYR1*-associated diseases may indicate the necessity to consider rhabdomyosarcoma. This association requires a genome-wide test such as exome sequencing searching for contributing variants in cancer-predisposing genes.

## Figures and Tables

**Figure 1 genes-14-01360-f001:**
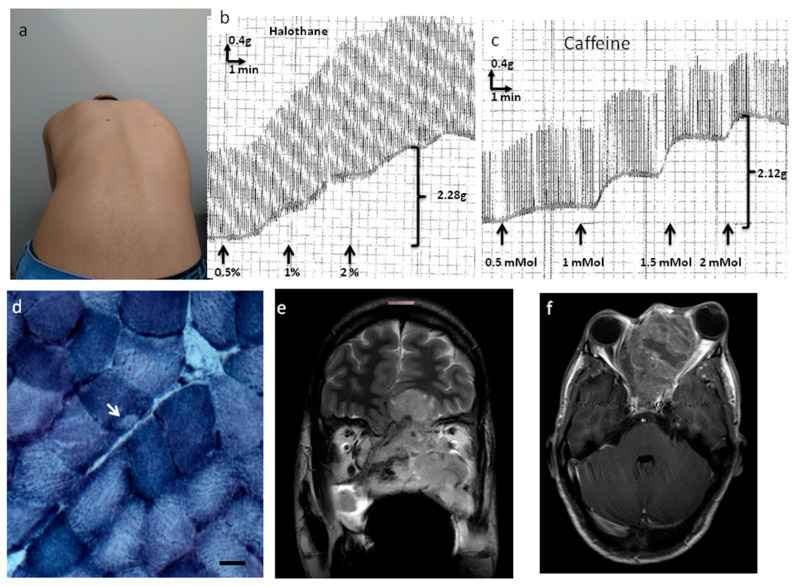
Clinical and laboratory findings: scoliosis, in vitro contracture test, skeletal muscle histochemistry, and magnetic resonance imaging. Legend: (**a**–**f**). (**a**) Positive Adam’s test for left gibbousness. (**b**,**c**) Positive in vitro contracture test for halothane (**b**) and caffeine (**c**). (**d**) Cross-sectional section of frozen skeletal muscle showing a core (white arrow) inside a type I muscle fiber (NADH, 100×, scale bar: 30 μm, Nikon Eclipse 50i/Nikon DS-Fi1 microscope, Tokyo, Japan). (**e**,**f**) Magnetic resonance images of the face and neck showing a solid, heterogeneous mass with ill-defined boundaries and infiltrative aspects (left: coronal T2 weight, right: T1 weight SE with gadolinium).

**Figure 2 genes-14-01360-f002:**
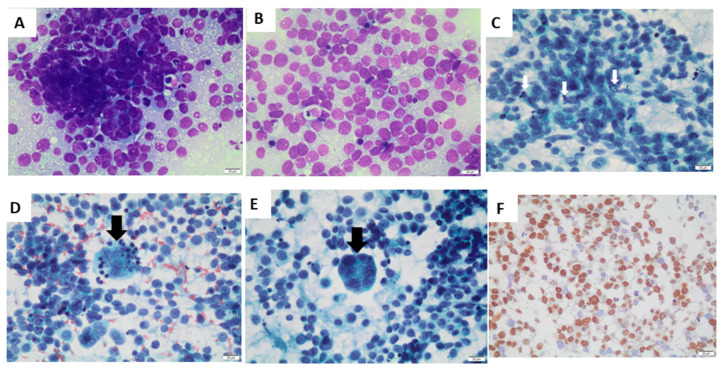
Cytologic features of cervical nodule fine-needle aspiration cytology. Legend: (**A**): Small, rounded cells, grouped or isolated (Giemsa stain). (**B**): Scarce and micro-vacuolated cytoplasm (Giemsa stain). (**C**): Mitosis (arrow; Papanicolaou stain). (**D**): Apoptotic bodies (arrow; Papanicolaou stain). (**E**): Multinucleated giant tumor cell (arrow; Papanicolaou stain). (**F**): Immunohistochemistry positive for myogenin. Bar scale: 20 μm, 400×, Olympus BX50/Olympus SC100 microscope.

**Figure 3 genes-14-01360-f003:**
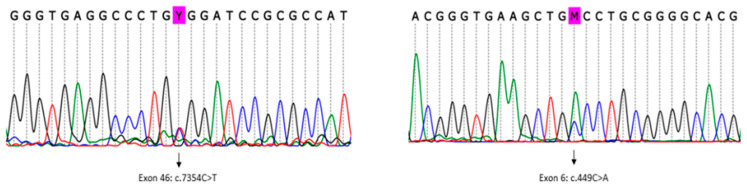
Germline *RYR1* and *ASPSCR1* missense variants were detected by whole-exome sequencing in the patient and validated by Sanger sequencing. Legend: Sanger sequencing results. (**left**) A heterozygous missense variant was detected in exon 46 of the gene *RYR1* (c.7354C>T). The variant was previously described as pathogenic in ClinVar (RCV000056226.1) and in the literature [15,16]. (**right**) A heterozygous missense variant was detected in exon 6 of the *ASPSCR1* gene (c.449C>A). The variant was classified as a VUS according to the ACMG criteria. The *ASPSCR1* gene has been associated with alveolar soft part sarcoma (OMIM: 606236).

## Data Availability

All data generated or analyzed during this study are included in this published article (and its Appendix A).

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
