# Peer review of "Rhabdomyosarcoma Associated with Core Myopathy/Malignant Hyperthermia: Combined Effect of Germline Variants in RYR1 and ASPSCR1 May Play a Role"

_genes, 2023, doi:10.3390/genes14071360_

Round 1
Reviewer 1 Report
Summary
This manuscript reports the case of the rhabdomyosarcoma patients with germline variants in the ryanodine receptor 1 (RYR1) and alveolar soft part sarcoma chromosomal region candidate 1 (ASPSCR1) genes. This is the valuable case report presenting the findings of rhabdomyosarcoma with malignant hyperthermia.
Comments
1. Lines 151-154: The results of the immunohistochemistry performed by the authors must be presented. Or, if it has been reported, the reference must be cited.
2. The titles of Sections 3.1, 3.2, and 3.3 should be the simple abstracts of the results.
Minor points
3. Line 19: “a 18-year-old” should be “an 18-year-old”.
4. Line 24: “ASPSCR1” should be defined in the Abstract.
5. Line 39, 57, 59: The gene names “RYR1” and “ASPSCR1” should be in italics.
6. Line 123: “ECG” should be defined.
The manuscript contains several grammatical and minor errors.
Author Response
Thank you for the very good comments and suggestions. See our corrections bellow:
Comments
- Lines 151-154: The results of the immunohistochemistry performed by the authors must be presented. Or, if it has been reported, the reference must be cited.
Detailed Data on the immunohistochemistry panel of the tumor are presented material and methods, and illustrated in figures 2f and supplemental figure 1.
- The titles of Sections 3.1, 3.2, and 3.3 should be the simple abstracts of the results.
Very good suggestion. The titles were changed accordingly.
Minor points
- Line 19: “a 18-year-old” should be “an 18-year-old”.
It was corrected.
- Line 24: “ASPSCR1” should be defined in the Abstract. Done.
- Line 39, 57, 59: The gene names “RYR1” and “ASPSCR1” should be in italics.
Thank you. All the names of the genes were corrected.
- Line 123: “ECG” should be defined. Ok.
Comments on the Quality of English Language
The manuscript contains several grammatical and minor errors.
A new revision of the English was done.
Reviewer 2 Report
The authors are asked to complete the following results and problems:
1. specifying exactly the panel of mutations tested in the patient 2. confirmation that mutations of TP53, BRCA2, CDNK2A and other mutations typical of Lynch syndrome, for example, have been excluded 3. opinions and consultations of the geneticist - the patient manifested many features of dysmorphia 4. providing information on whether NGS testing was performed in the muscle tissue that was biopsied 5. Please describe the involvement of RYR1 and ASPSCR1 somatic mutations in cancer - these data are well described in the literature.
The paper is interesting and shows the potential involvement of downstream genes in cancerogenesis.
Author Response
We could like to thank the reviewer for the very good comments, that were all included in the manuscript.
The authors are asked to complete the following results and problems:
- specifying exactly the panel of mutations tested in the patient
This information was described in the methodology; it was performed whole-exome sequencing (“Germline whole-exome sequencing was performed with blood DNA to construct genomic libraries using the Sure Select Human All Exon V6 kit (Agilent Technologies)). Therefore, all known gene related to diseases could be checked for variants.
- confirmation that mutations of TP53, BRCA2, CDNK2A and other mutations typical of Lynch syndrome, for example, have been excluded
All variants mapped to known (including TP53, BRCA2, CDNK2A) or candidate cancer predisposition genes were evaluated. The patient did not carry any clinically relevant variant (pathogenic/likely pathogenic) in cancer predisposition genes besides the one mentioned in the manuscript (ASPSCR1). This information was now added to the manuscript.
- opinions and consultations of the geneticist - the patient manifested many features of dysmorphia
We thank the Reviewer for this observation. The facial dysmorphisms exhibited by the patient were rather unspecific and a clinical hypothesis could not be made. In spite of these facial dysmorphisms, additional clinically relevant variants in OMIM morbid genes were not detected. However, as already discussed in the manuscript, the patient present features that could be found in recessive RYR1 mutations, such as facial dysmorphism, strabismus/ptosis, joint hyperlaxity, muscle hypotrophy and increased nuclear centralization. Previous reports of the pathogenic variant c.7354C>T in RYR1 described a variable pattern of dysmorphic features. These remarks were included in the manuscript.
- providing information on whether NGS testing was performed in the muscle tissue that was biopsied
Whole-exome sequencing and CMA were performed using DNA extracted from peripheral blood.
- Please describe the involvement of RYR1 and ASPSCR1 somatic mutations in cancer - these data are well described in the literature.
We thank the Reviewer for this comment. We expanded the discussion regarding this issue in the manuscript (please find below).
“Genetic predisposition was detected in only 7-10% of rhabdomyosarcoma patients [15]; in the present case, mutations in known cancer susceptibility genes could be excluded, but notably, a missense VUS was detected in ASPSCR1, and such variants have been associated with glioma and leukemia in pediatric patients [15]. Some cases of alveolar soft tissue sarcomas have been found to be caused by fusion of ASPSCR1 and TFE3 [16]. In addition, recently it was shown that in alveolar soft part sarcoma, the expression of the fusion transcription factor ASPSCR1::TFE3 is required for in vivo tumor development via angiogenesis, orchestrating higher ordered angiogenesis via super enhancers activity (https://pubmed.ncbi.nlm.nih.gov/37029109/). The protein encoded by ASPSCR1 belongs to the family of UBX domain-containing proteins and regulates the trafficking of GLUT4 glucose transporters in adipocytes and muscle cells [17,18]. Since the Golgi multicompartment stack structure provides an optimal setting for glycosylation, dysregulation of the Golgi structure and Golgi structural proteins has been highly related to glycosylation defects [19]. The role of the Golgi complex and glycosylation defects in cancer has been recently highlighted [20]. Therefore, it is legitimate to speculate that a germline deleterious variant in genes causing Golgi dysregulation, such as ASPSCR1, may lead to altered glycosylation and possibly contribute to tumorigenesis in cells expressing such mutant or detective protein, such as adipocytes and muscle cells. Taken together, these pieces of evidence imply a possible role for the detected germline ASPSCR1 variant in rhabdomyosarcoma development.
Other relevant issue to be considered is the role of the germline RYR1 variant in the cancer development in this case. Abnormalities in Ca2+ signaling are frequently observed in tumors, representing common alterations (https://pubmed.ncbi.nlm.nih.gov/28386091/). The remodeling of Ca2+ signaling in cancer can have significant impacts on crucial processes such as proliferation, invasion, and cellular susceptibility to death. Among the various types of Ca2+ release channels, one notable example is RYR, and the mutational landscape of the three RYR isoforms have not been systematically investigated until a recent study by Wang et al (2022). In this study (https://pubmed.ncbi.nlm.nih.gov/36167878/), they evaluate the mutational landscape of RYR and its correlation with various cancer phenotypes, utilizing data from the TCGA database; their findings unveiled characteristic RYR mutational patterns in cancers and established correlations with tumor mutational burden (TMB), patient prognosis, age, and smoking status; however, rhabdomyosarcomas were not evaluated. So far, there is no reported link between RYR1 gene mutations and rhabdomyosarcoma development. However, similar to dystrophin deficiency, increased intracellular calcium, a characteristic of many myopathies [1], could be theoretically a contributing factor in patients who already carry variants in genes that may increase rhabdomyosarcoma susceptibility. SR calcium-ATPase (SERCA) is expressed in rhabdomyosarcomas, suggesting that intracellular calcium could participate in tumor cell proliferation [21]. Additionally, CD97 (an adhesion class G-protein-coupled receptor), which is expressed at the peripheral sarcolemma and colocalizes with SERCA, is strongly upregulated in rhabdomyosarcomas compared to normal skeletal muscle [22] and could be involved in myogenic differentiation of sarcomas.
The paper is interesting and shows the potential involvement of downstream genes in cancerogenesis.
Thank you very much!
Round 2
Reviewer 2 Report
Dear Authors,
Thank you for the remarks introduced, I have no further comments